# Psychology in action: Social media communication, CSR, and consumer behavior management in banking

Yang Liu[1], Rana Tahir Naveed[2]*, Sara Kanwal[3,4], Muhammad Tahir Khan[2], Ali F. Dalain[5], Wei Lan[6]

1 School of literature and journalism, Xihua University, Sichua, Chengdu, China, 2 Division of Management and Administrative Sciences, University of Education (UE) Business School, University of Education, Lahore, Pakistan, 3 Institute of Business and Management (IB&M), University of Engineering and Technology (UET), Lahore, Pakistan, 4 Graduate School of Business (GSB), Universiti Kebangsaan Malaysia (UKM), Selangor, Malaysia, 5 Department of Human resource Management, College of Business Administration, University of Jeddah, Jeddah, Saudi Arabia, 6 Chongqing Vocational Institute of Engineering, Chongqing, China

☉ These authors contributed equally to this work.
* tahir.naveed@ue.edu.pk

**Data Availability Statement:** All relevant data are within the paper and its Supporting Information files.

## Abstract

In today's digitally interconnected world, social media emerges as a powerful tool, offering different opportunities for modern businesses. Not only do organizations use social media for marketing purposes, but they also endeavor to influence consumer psychology and behavior. Although prior studies indicate social media's efficacy in disseminating corporate social responsibility (CSR) communications, there remains a dearth of research addressing the impact of CSR-related messaging from banks on consumers' brand advocacy behavior (CBAB). Our study seeks to bridge this gap, exploring the CSR-CBAB relationship within the banking sector of an emerging economy. Additionally, we investigate the roles of consumers' emotions and values in mediating and moderating their CBAB, introducing two mediating factors, consumer happiness (HP) and admiration (BRAD), and moderating variable altruistic values (ATVL). Data collection involved an adapted questionnaire targeting banking consumers. The structural analysis revealed a positive correlation between a bank's CSR-related social media communications and CBAB. HP and BRAD were identified as mediators in this relationship, while ATVL emerged as a moderator. These findings hold significant theoretical and practical implications. For instance, our research highlights the indispensable role of social media in effectively conveying CSR-related information to banking consumers, subsequently enhancing their advocacy intentions.

## 1. Introduction

Establishing consumer relationships has been vital for enduring business success. The present global economic turmoil, waning traditional media efficacy, and consumer brand skepticism necessitate further emphasis on nurturing consumer-brand relationships [1]. Research

**Funding:** The author(s) received no specific funding for this work.

**Competing interests:** The authors have declared that no competing interests exist.

highlights that consumer-focused cultures yield superior organizational outcomes [2, 3], enriching marketing literature and illuminating how loyal consumers bolster a company's achievements [4, 5]. Consequently, fostering meaningful, long-term consumer relationships is paramount in today's digital age [6, 7]. Recent literature accentuates the significance of various communicative behaviors, such as positive word-of-mouth [8, 9], product and service referrals [10, 11], and brand endorsements [12, 13], in shaping consumer-brand relationships. Consumers prefer peer recommendations over marketing advertisements during purchasing decisions. Nielsen's data illustrates that approximately 90% of consumers rely on recommendations from their social circles [14], indicating greater trust in peers than in company-originated messages [15]. While the literature acknowledges the role of consumers' communicative behavior, companies face challenges in promoting specific behaviors, as capturing consumers' attention remains elusive [16, 17]. Thus, businesses must carefully strategize to influence consumer communicative behavioral intentions, impacting brand reputation and satisfaction.

The digital era has rendered consumers increasingly informed, empowered, and connected [18, 19], necessitating brands to prioritize transparency, authenticity, and trustworthiness to maintain loyalty and satisfaction [20, 21]. Utilizing digital strategies, such as social media, can enhance consumer engagement and relationship-building [22]. The advent of the internet and digital technologies, particularly social media, has heightened consumer connectivity, interactivity, and awareness [23], pressuring companies to adopt transparent and responsible practices [24, 25].

Corporate Social Responsibility (CSR) has become an essential business strategy in recent years [26–28], holding companies accountable for social and environmental performance [29, 30]. CSR initiatives, including sustainability efforts like reducing carbon emissions and utilizing renewable energy, are now integral to business strategy [31]. CSR-based actions contribute to building positive public opinion and trust, strengthening a brand's image. Brands' interest groups, including consumers, are keen to learn about their CSR strategies to preserve the biosphere and communities [32]. Companies like Starbucks and Patagonia exemplify a commitment to CSR initiatives [33, 34]. In this vein, contemporary consumers consider environmental impact when evaluating products or services [35], leading to a focus on sustainable offerings and companies striving to reduce their carbon footprint. This has spurred the growth of carbon offset services, enabling consumers to counterbalance emissions through sustainable project investments [36]. Consequently, a brand's CSR engagement holds increased importance for modern consumers [37].

Social media has transformed businesses, enabling direct consumer connections and data-driven insights for product refinement and targeted marketing. Platforms like Instagram, Facebook, and Twitter facilitate personalized experiences, increasing consumer engagement and sales [38]. Brands can use social media to inform consumers and other stake holders about CSR initiatives [39, 40]. However, academic literature highlights consumers' preference for peer recommendations over company marketing [41]. Consumers trust peers more due to perceived honesty, relatability, and shared experiences [42].

This study examines how CSR-related information by a particular brand on different social media platforms influences consumers' advocacy behavior (CBAB). Indeed, CBAB occurs when consumers promote or defend a brand for specific reasons [43]. Although research suggests CSR influences consumer behavior, some critical gaps in the existing body of knowledge remain [44, 45], which this study aims to address. Firstly, CSR scholars often study how a company's CSR actions influence consumer behavior, including loyalty intentions [46, 47] and brand preferences [48]. Consumers tend to prefer socially responsible companies and show more loyalty toward them. While these studies provide valuable insights into consumer

psychology, we propose that the CSR- CBAB relationship deserves more attention due to consumer recommendations and endorsements' significance in decision-making. Secondly, emotions play a crucial role in human psychology [49], with the average person experiencing over 400 emotional events daily [50]. Emotions greatly influence consumers' purchasing decisions, as they associate feelings with specific brands. Despite this, many CSR studies have adopted rational approaches [51, 52], with critiques arguing for the importance of emotions in decision-making [53, 54]. We aim to fill this gap by incorporating consumers' happiness (HP) and brand admiration (BRAD) as mediators in the CSR- CBAB relationship. The inclusion of these variables as mediators in this study lies with logic provided by various scholars who emphasized the importance of different mediators in a CSR framework to influence individual behavior [55, 56]. Servera-Francés and Piqueras-Tomás [57] also mentioned that CSR influences consumer behavior via different variables as mediator(s). Therefore, this study proposes HP and BRAD as mediators to explain the underlying mechanism of the CSR and CBAB relationship.

Similarly, human values guide behavior and serve as motivation, with individuals more likely to act on values [58–61]. Nonetheless, human values need specific contexts to manifest a particular behavior [62, 63]. It has recently been debated how human values can moderate a particular attitude or behavior [64, 65]. To this end, we propose altruistic values (ATVL) in a CSR context that can moderate the CSR- CBAB mechanism via HP and BRAD. Thirdly, this study contributes to CSR-consumer behavior research in developing countries like Pakistan, as CSR is culturally and contextually specific [66], making direct comparisons with developed countries difficult. Lastly, the traditional consumer relationship model (CRM) faces criticism for being inefficient in establishing long-term consumer-company relationships [67, 68]. Actually, the conventional CRM model often relies on short-term economic transactions and lacks personalization. Companies should leverage technology, for example, social media, for more meaningful engagements and two-way communication strategies. Still, social media's role in CSR communication and consumer relationship management remains underexplored.

Promoting CBAB offers numerous advantages for organizations across industries and sizes. However, having consumers as brand advocates becomes even more critical in the service industry context. This is because, in service settings, prior experience and testing are not likely [69–71]. Consequently, CBAB holds particular significance for the services sector. Consumer advocacy is crucial for the banking sector, as it can enhance trust and reputation. With limited differentiation opportunities, banks rely on consumer experiences and word-of-mouth to attract new clients. Positive CBAB can lead to increased consumer loyalty and retention, further impacting a bank's market position. In the context of the banking industry, banks that aptly articulate and fulfill their CSR obligations receive positive customer evaluations. Initiatives such as backing renewable energy projects, aiding local enterprises, or introducing financial literacy programs gain considerable favor with consumers. Successful examples of such practices include banks like Bank of America, which pledged $300 billion towards low-carbon business efforts, and Standard Chartered, renowned for its numerous community engagement initiatives. Highlighting a specific example from Pakistan, Habib Bank Limited (HBL) has demonstrated exceptional success with its comprehensive CSR programs. HBL has made significant inroads in endorsing education, public health, and environmental sustainability, thereby positively influencing its client base. In a similar vein, United Bank Limited (UBL) has put forth considerable CSR contributions, with an emphasis on community development and disaster relief. These CSR endeavors solidify the banks' image and motivate customers to champion these brands, leading to enhanced loyalty and customer retention. Hence, by fostering CBAB through CSR initiatives, banks can secure a competitive advantage and drive long-term success. In addition to trust and reputation, strong CAB in the banking sector can reduce

consumer acquisition costs, as satisfied clients become brand ambassadors. Furthermore, CBAB can help banks navigate crises, as loyal advocates provide a supportive base during turbulent times. Moreover, emphasizing CBAB through CSR initiatives also aligns banks with societal expectations, potentially influencing regulatory environments in their favor. Thus, investing in CBAB is of paramount importance for banks' overall performance and sustainability.

Specifically, our study tends to highlight the profound influence of CSR activities on consumer advocacy because the past literature suggests that CSR can influence consumer behavior significantly [72, 73]. When brands not only engage in, but also effectively communicate their CSR initiatives through social media, it stimulates consumers to advocate and defend the brand online, thereby enhancing the brand's reputation and trustworthiness. Furthermore, our research tends to demonstrate the crucial mediating role of HP and BRAD in the relationship between CSR and CBAB. CSR initiatives that promote HP and BRAD subsequently amplify CBAB. This suggests that by strategically aligning CSR activities to generate positive emotions among consumers, brands can substantially enhance the advocacy intentions of consumers.

Similarly, the proposed findings of this research will be helpful for brands in acknowledging the role of ATVL as a significant moderating factor in the CSR-CBAB mechanism. We believe that individuals with strong altruism are more likely to express CBAB when brands actively participate in meaningful CSR activities. Consequently, brands can design their CSR strategies and communications to cater to consumers' altruistic tendencies, fostering stronger consumer-brand relationships and ultimately fostering their advocacy intentions. More specifically, in the context of the banking sector, our research tends to indicate that CSR-induced CBAB significantly reinforces trust and reputation. In an industry with limited differentiating opportunities, CSR initiatives, by enhancing CBAB, appear to have substantial implications for client acquisition and retention. Furthermore, we propose that a strong CBAB can provide a supportive base during turbulent times, aiding banks in crisis management. Moreover, using social media platforms to disseminate information about CSR initiatives greatly enhances bank perceptions and advocacy. Hence, the proposed findings of this research will be helpful for brands to recognize the value of these platforms as they provide a perfect medium for two-way dialogue with consumers, thereby facilitating the building of stronger relationships.

## 1.1 Research objectives

The study aims to understand how CSR activities by a brand influence consumers' advocacy behavior.

The study seeks to investigate the role of HP and BRAD as mediators in the relationship between CSR and consumer advocacy.

The study aims to determine whether ATVL can moderate the influence of CSR on consumer advocacy through the mediating variables of HP and BRAD.

The study plans to contribute to CSR-consumer behavior research in developing countries like Pakistan, acknowledging that CSR can be culturally and contextually specific.

The study intends to look specifically at the banking industry and how CSR-induced consumer advocacy can reinforce trust, reputation, and customer retention in this sector.

The study aims to recognize the role of social media platforms in disseminating information about CSR initiatives and enhancing consumer-brand relationships.

The study aim to fill gaps in the current body of knowledge related to how CSR impacts consumer behavior and the need for emotional considerations in these studies.

This research comprises four sections: literature review, methodology, results, and discussion. The literature review presents relevant studies and hypotheses, while the methodology details data collection processes. The results section covers statistical tests and hypothesis testing. Finally, the discussion compares findings to prior research, discusses implications and limitations, and concludes the study.

## 2. Literature

Using social identity theory (SIT), we explore the relationship between CSR and CBAB. SIT suggests that social groups, such as banking organizations, can influence individuals' self-concept [74]. People develop strong associations with groups that share their values, leading them to adopt group beliefs and attitudes for approval and acceptance [75, 76]. CSR activities can prompt consumers to identify with organizations due to their ethical commitments. SIT has been widely used to study human behaviors in response to CSR activities [77, 78] and marketing outcomes [79, 80]. We propose that SIT can explain consumer advocacy intentions for a particular banking organization practicing CSR.

CSR positively influences consumer attitudes, impacting loyalty, purchase likelihood, citizenship behavior, and satisfaction [81, 82]. It encourages extra-role behaviors [83–85], including communicative behavior, where consumers recommend ethical companies to others [86, 87]. The digital age has enhanced company-consumer interactions through social media [88], enabling feedback, product research, and information sharing [89]. Ahmad, Naveed [40] and Chu and Chen [90] emphasize social media's role in improving CSR communication, consumer loyalty, and positive word-of-mouth. Well-planned CSR activities influence CBAB, as consumers share positive experiences and trust socially responsible organizations, leading to loyalty, repeat purchases, and ethical context sharing on social media. Therefore,

**H1:** Using social media for CSR communication can positively predict CBAB

Happiness has historically been considered the "highest good" [91], and marketers increasingly focus on consumer happiness for successful brand management [92, 93]. Consumers prefer brands offering emotional experiences [94], and brands connecting emotionally with consumers are more successful [95]. Understanding consumers' emotions toward brands helps marketers create targeted campaigns [96].

HP, an emotional response toward brands [97], warrants attention. Companies incorporating CSR into their communication strategies adapt better [98]. Socially responsible acts of a business and product/service provenance are increasingly important to consumers [99]. Brands' CSR communication with stakeholders can connect with diverse audiences [100]. Hamid, Riaz [101] found CSR strategies led to positive social construction and brand development. Happiness Management introduced CSR to foster positive consumer feelings [100, 102]. Adib, Zhang [103] found CSR can increase brand identification and positive emotions. While CSR and product price may have an in-direct correlation [104], emotions can mediate the relationship. Previous studies have discussed the mediating role of emotions [105, 106]. Gupta, Nawaz [39] showed that CSR communication enhances purchase intentions while emotions mediate this relationship. Thus, we propose:

**H2:** CSR communication on social media by a particular bank positively predicts HP

**H3:** HP mediates between CSR-related communication on social media and CBABCBAB

CSR-oriented organizations gain relational advantages as consumers feel prestige from purchasing ethical organization [107]. Such organizations engage consumers emotionally [108] and develop positive emotions like admiration. Emotions' role in shaping consumer behavior is well-documented [109, 110], and CSR fosters positive emotions [87, 111]. Fortune 500's report shows admired brands are socially responsible [112]. Companies like Google, Disney,

and Microsoft are renowned for quality and ethical practices. CSR's positive link to consumer emotions, for example BRAD, is established in academic literature [39, 113].

Strong consumer emotions resulting from CSR drive CBAB [108]. Emotional attachment to a brand creates strong bonds, and emotions influence advocacy [114]. A Harvard Business Review report highlights the benefits of emotional connections with consumers [115]. From SIT standpoint, a bank's CSR activities establish admiration and foster a positive image, attracting responsible consumers. CSR creates consumer pride and loyalty, leading to increased advocacy. Given emotions' potential to drive advocacy, we propose:

**H4:** A positive association exists between CSR and BRAD

**H5:** BRAD mediates between CSR and CBAB

Emotions and values are intrinsically linked, influencing each other [116]. They are essential for decision-making and guiding behavior [117]. Appraisal theories of emotion suggest value concerns cause emotions [118]. Emotions intensify when related to individuals' values [119], leading to increased emotional investment [135]. Personal values shape individual choices, decisions, responses, and actions. The "empathy-altruism hypothesis" [120] posits that those valuing compassion, including benevolence and universalism, exhibit altruism, empathy, and happiness [121]. The moderating role of personal values, like ATVL, in influencing extra-role behavior has been emphasized [122]. Guan, Ahmad [26] validated ATVL's conditional indirect role in shaping extra-role behavior within a CSR framework. Bigné-Alcañiz, Currás-Pérez [123] and Romani, Grappi [106] confirmed AV's moderating function in consumer-brand relationships and buffering consumer emotions influencing extra-role behavior. As CBAB is an extra-role behavior and personal values buffer the relationship between emotions and extra-role behaviors, we propose the following:

**H6:** ATVL moderates the path between CSR and CBAB mediated by HP

**H7:** ATVL moderates the path between CSR and CBAB mediated by BRAD

## 3. Methods

### 3.1 Study sector and data collection

The banking services industry in Pakistan was chosen as the focus of this study for two key reasons. Firstly, relative to developed nations, CSR is an evolving concept in Pakistan, where many economic sectors lack a mature CSR framework [79]. Despite this, the banking industry stands out as one of the most significant and innovative sectors, boasting an established CSR structure [124]. Pakistani banks engage in a range of CSR initiatives, such as philanthropy, supporting the disadvantaged, community education, sustainability, and more. Given its mature CSR framework, large customer base, and standardized operations, this sector presents a valuable focus for our research. Secondly, despite possessing a well-defined CSR structure, Pakistani banks actively leverage social media to engage with customers and showcase their CSR efforts [39]. Many banks utilize digital platforms for organizational communication with external communities, including customers. This communication often involves sharing the bank's CSR efforts for societal and community benefit. Major banks maintain a presence on social media platforms where they engage in dialogue with customers, using these spaces not only for promotional activities but also to foster CSR-related discussions [40].

For our data collection, we selected six leading banks in Pakistan (four conventional and two Islamic). Data were gathered from banking customers in Lahore, the provincial capital of Punjab. We used a purpose-built data collection strategy, focusing on customers as they left bank branches or used ATM services. This approach offered two main advantages: direct interaction with actual banking customers and no disruption to the delivery of banking services. Similar data collection methods have been employed by CSR researchers like Ahmed, Zehou [125] and

Sun, Rabbani [124] in the context of banking services. We approached customers at various times of the day, asking them to fill out questionnaires. Screening questions ensured respondents had a basic understanding of CSR, and informed consent forms were provided to assure voluntary participation [126, 127]. With regard to ethical considerations, we adhered to the key principles of the Helsinki Declaration, as recommended by other authors [128–131]. Data collection was carried out from banking customers during November 2022 and January 2023.

## 3.2 Instrument

Utilizing a paper-pencil approach, we presented banking consumers with a 5-point Likert scale questionnaire, adapting items from reliable sources. Five CSR items were derived from Fatma, Ruiz [132], modifying Brown and Dacin [133] and Klein and Dawar [134] scales for social media. A sample item read, "When I see CSR-related communication of my bank on a social networking website, I feel that this bank is socially responsible." For HP, we used five items from Fei, Zeng [135], with a sample item being "buying the services of a socially responsible bank (like this) makes me happy." BRAD was gauged using Sweetman, Spears [136] five-item scale; a sample item being, "I feel admiration when I think about this banking organization." Melancon, Noble [137] provided the scale for CBAB (four items), with a sample statement being, "I would defend this bank on social media to others if I heard someone speaking poorly about it." Eight ATVL items were adapted from Schwartz [138] scale, with one item reading, "As a guiding principle in my life, I consider working for the welfare of others."

The questionnaire's statements were evaluated by academic and banking experts, and the final version was distributed to consumers upon their endorsement. Anticipating a low response rate, we initially distributed 600 questionnaires. Ultimately, we collected 409 filled questionnaires. After removing 19 invalid cases due to missing information, we identified 390 responses as valid. We distributed the questionnaires directly to customers, engaging them as they exited bank branches or made use of ATM services in densely populated areas of Lahore. This hands-on approach allowed us to instantly address any questions participants might have had, thereby enhancing the credibility of the data collected. Moreover, it ensured that our data collection process did not intrude upon regular banking services. In terms of the ATM locations, we pinpointed areas with high customer traffic in the vicinity of the chosen banks. This strategy was adopted to maximize our chances of interfacing with a diverse customer population that regularly interacts with the banks' services.

## 3.3 Social desirability and common method bias

Various theoretical steps were taken to minimize social desirability and common method bias (CMB), such as anonymous surveys and randomized response techniques. With these approaches, participants remained unaware of their specific survey involvement or response. Consequently, they had no motivation to provide answers that portrayed them favorably, as their input could not be traced back to them. Moreover, we clarified to respondents that only sincere responses, without right or wrong, were needed. Additionally, the questionnaire avoided vague statements and minimized respondent fatigue by using short-scale versions. These shorter scales demanded less time and effort, ensuring meaningful responses.

Empirical CMB detection was checked by using Harman's single-factor test [139], revealing a single factor responsible for about 30% of the variance, which is less than the 50% threshold. This finding implies that the variance stems from diverse factors within the data, rather than one dominant aspect. Thus, no CMB issue was present. Out of the total participants, males accounted for a majority share of 67.18% (262 individuals), whereas females constituted 32.82% (128 individuals). In terms of age demographics, the respondents were divided into

five different age groups. The age group with the highest representation was the 26–35 years old bracket, comprising 36.92% of the total respondents. Following this, the age groups of 36–45 and 46–50 years represented 20.51% and 17.70%, respectively. The younger age group of 18–25 years represented a lesser portion, contributing to 11.79%, while the respondents aged above 50 years accounted for the remaining 13.08%. Regarding the educational background of respondents, those holding a graduate degree made up 37.95% (148 individuals), and 35.13% (137 individuals) possessed a master's degree. Respondents with an intermediate level of education constituted a smaller segment of 12.56% (49 individuals). Those who had an education beyond the master's degree represented 14.36% (56 individuals) of the total respondents.

## 4. Results

The study's data were analyzed through factor analysis on outer items, a method proficient in distinguishing variable structures and their inter-relationships. By examining factor loadings, or the impact of each item on the variable, we confirmed that 24 outer items out of a total of 27 significantly influenced their respective variables, with loadings above the 0.5 threshold. In this respect, we removed two items of ATVL and one item of CBAB due to their week factor loadings. Convergent validity was evaluated by examining the average variance extracted (AVE), providing insight into each variable's explanatory power. The AVE ranged between 0.556 and 0.730 for ATVL and CBAB, as detailed in Table 1 and Fig 1.

Reliability testing of each variable used Cronbach alpha, Rho_A, and composite reliability values, assessing internal consistency and overall reliability, respectively. All values surpassed the significant 0.7 threshold, exemplified by ATVL's results of 0.839, 0.842, and 0.882 for Cronbach alpha, Rho_A, and composite reliability, respectively (Table 2).

Correlation analysis showed significant, positive inter-variable relationships, suggesting potential causal connections. ATVL and BRAD had a correlation of 0.547, while ATVL and CBAB showed a value of 0.331. Discriminant validity tests ensured that a variable's items were unique from all other variables (Table 3). Additionally, Hetrorait-Monotrait Ratio (HTMT) results also confirmed that the discriminant validity was significant in all cases (Table 4).

By using structural equation modeling in SMART-PLS software, we tested our hypothesized relationships. This advanced tool, designed for complex data analysis [140–142], confirmed all direct relationships were significant. It validated hypotheses H1, H2, and H4, establishing direct, significant relationships between CSR and CBAB, CSR and HP, and CSR and BRAD. Significant indirect effects were also discovered, with partial mediation between CSR and CBAB via HP and BRAD, implying these mediating factors play a significant role in CSR's impact on CBAB. This validated that H3 and H5 were statistically significant. Furthermore, the interaction term (CSR x ATVL) held a statistically significant moderating effect on the mediated relationship between CSR and CBAB via HP and BRAD. This confirmed that there is a significant conditional indirect effect of ATVL on the above relationship. Hence H6 and H7 were also significant. Table 5 and Fig 2 summarize these results.

## 5. Discussion

Empirical analysis reveals that the CSR communication of banking organizations on social networking websites substantially predicts banking consumers' advocacy intentions. This is in line with previous researchers [90, 143, 144]. This implies that organizations can elevate consumers' psychology and behavior intentions by sharing CSR activities on social media, fostering support for socially responsible banking organizations. Consumers appreciate and widely share CSR-related information on social media, transforming into advocates and positively impacting the organization's reputation. In the digital age, consumers increasingly consider

**Table 1. Outer loadings and validity.**

|  | λ | S.D | T.Values | AVE |
|---|---|---|---|---|
| **ATVL** |  |  |  | 0.556 |
| ATVL_2 ← ATVL | 0.690 | 0.038 | 18.1 |  |
| ATVL_3 ←ATVL | 0.750 | 0.034 | 22.369 |  |
| ATVL_4 ← ATVL | 0.746 | 0.032 | 23.226 |  |
| ATVL_5 ← ATVL | 0.815 | 0.022 | 37.614 |  |
| ATVL_6 ← ATVL | 0.729 | 0.034 | 21.326 |  |
| ATVL_7 ← ATVL | 0.737 | 0.034 | 21.642 |  |
| **BRAD** |  |  |  | 0.631 |
| BRAD_1 ← BRAD | 0.798 | 0.028 | 28.199 |  |
| BRAD_2 ← BRAD | 0.786 | 0.027 | 29.026 |  |
| BRAD_3 ← BRAD | 0.838 | 0.02 | 40.986 |  |
| BRAD_4 ← BRAD | 0.728 | 0.041 | 17.936 |  |
| BRAD_5 ←BRAD | 0.818 | 0.021 | 38.125 |  |
| **CBAB** |  |  |  | 0.730 |
| CBAB_1 ← CBAB | 0.840 | 0.023 | 36.469 |  |
| CBAB_2 ← CBAB | 0.859 | 0.023 | 37.614 |  |
| CBAB_3 ← CBAB | 0.865 | 0.018 | 47.927 |  |
| **CSR** |  |  |  | 0.632 |
| CSR_1 ← CSR | 0.757 | 0.029 | 26.197 |  |
| CSR_2 ← CSR | 0.847 | 0.018 | 45.957 |  |
| CSR_3←CSR | 0.809 | 0.021 | 39.296 |  |
| CSR_4 ← CSR | 0.732 | 0.035 | 20.685 |  |
| CSR_5 ← CSR | 0.827 | 0.022 | 38.007 |  |
| **HP** |  |  |  |  |
| HP_1 ← HP | 0.782 | 0.027 | 29.463 | 0.613 |
| HP_2 ← HP | 0.814 | 0.024 | 34.448 |  |
| HP_3 ← HP | 0.796 | 0.025 | 32.029 |  |
| HP_4←HP | 0.753 | 0.029 | 25.753 |  |
| HP_5 ←HP | 0.770 | 0.027 | 28.978 |  |

**Notes:** λ = Item loadings, AVE = Average variance extracted

the environmental and social impacts of their purchase decisions. Social media platforms enable banking organizations to interact effectively with consumers and facilitate dialogical communication. CSR communications have been emphasized as a way to influence consumers' extra-role behaviors in the social media era. Transparent and public CSR messages on social media can evoke emotional connections with consumers, leading to loyalty and recommendations. From a social identity perspective, ethical banks' socially responsible behavior can encourage strong consumer identification. This can result in increased loyalty and long-term success. Our study verifies that CSR-related communication positively affects CBAB.

Our findings also highlight the significance of consumer emotion in shaping CBAB. Higher levels of positive emotions correspond to increased engagement in CBAB. Consequently, emotional states should be considered when designing CSR strategies. HP and BRAD significantly mediate the CSR-CBAB relationship, emphasizing the importance of generating positive consumer experiences on social media, particularly regarding CSR [105, 106, 145]. This can lead to increased consumer advocacy intentions, ultimately benefiting socially responsible organizations. To this end, results indicate that CSR positively influences HP and BRAD of

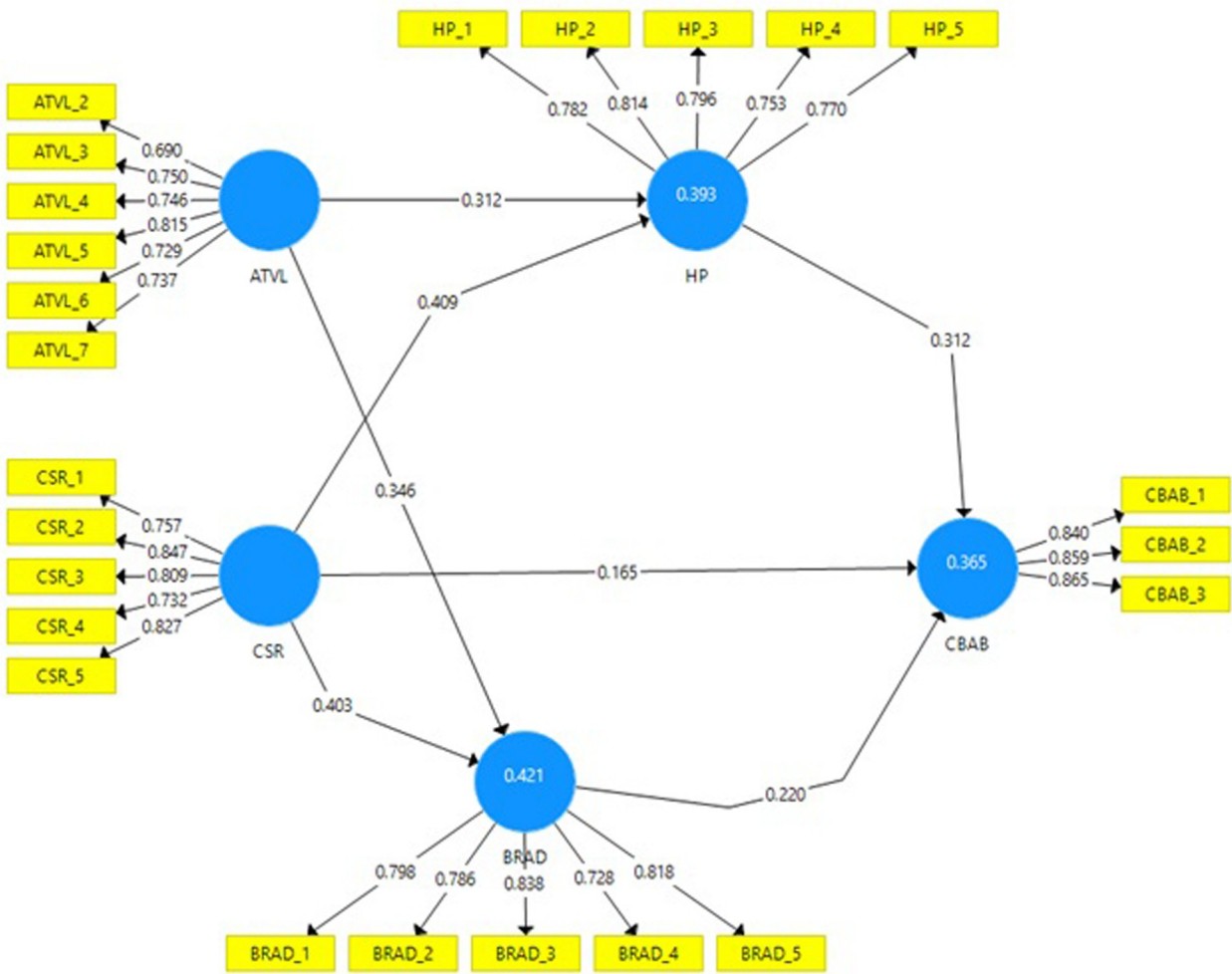

**Fig 1. The measurement model showing the factor loadings of items along with R-square values.**

consumers. These findings hold considerable importance for consumer relationship management strategies. Developing long-term emotional relationships is crucial in the digital age. Consumers develop emotional attachments with the banks that prioritize CSR strategies encompassing societal and ecosystem betterment. CSR enhances consumer emotional engagement, resulting in higher HP and BRAD levels for sustainable banking organizations and increasing trust and emotional engagement.

Lastly, AVTL served as a conditionally indirect moderator of CSR-CBAB through HP and BRAD. Altruism-driven personal values positively influence consumers' emotional state.

**Table 2. Reliability statistics.**

| Variable | Cronbach alpha | Rho_A | Composite reliability |
|---|---|---|---|
| ATVL | 0.839 | 0.842 | 0.882 |
| BRAD | 0.854 | 0.861 | 0.895 |
| CBAB | 0.815 | 0.816 | 0.89 |
| CSR | 0.854 | 0.86 | 0.896 |
| HP | 0.842 | 0.843 | 0.888 |

**Table 3. Correlations and discriminant validity.**

|  | ATVL | BRAD | CBAB | CSR | HP |
|---|---|---|---|---|---|
| ATVL | 0.745 |  |  |  |  |
| BRAD | 0.547 | 0.794 |  |  |  |
| CBAB | 0.331 | 0.525 | 0.855 |  |  |
| CSR | 0.499 | 0.575 | 0.468 | 0.795 |  |
| HP | 0.517 | 0.676 | 0.554 | 0.565 | 0.783 |

ATVL conditionally moderates the effect, depending on the CSR-CBAB relationship strength through HP and BRAD. Indeed, emotions intensify when related to an individual's needs, goals, motives, values, norms, and attachments. These core elements guide behavior and response to external stimuli. CSR actions by banks and consumers' ATVL create value congruence, buffering consumers' emotions (such as HP and BRAD). We confirm the conditional indirect effect of AV in this study. Moreover, our analysis identified several key factors within each variable that contribute significantly to CBAB. For instance, within CSR, specific aspects such as environmental responsibility and community outreach seem to have a more potent influence on CBAB. For HP, positive experiences related to personalized interaction and prompt service resolutions are particularly impactful. As for BRAD, aspects like trustworthiness, authenticity, and responsiveness of the brand may be critical in shaping CBAB. Lastly, within AVTL, consumers who value altruistic acts and ethical conduct are more likely to engage in CBAB. Table 6 represents a summary of the key findings.

## 5.1 Implications for theory

The current research contributes distinct theoretical insights to the existing literature, significantly enriching the knowledge base. First, previous CSR studies in consumers' psychology and behavior management primarily explored how CSR predicts behavioral intentions like loyalty [46, 47] or brand purchase intentions [48, 146]. However, research on CSR's influence CBAB remains scarce. CSR-CBAB relationships hold more importance than loyalty or brand preferences, as consumers trust peer recommendations and often purchase recommended products and services. These relationships foster trust between consumers and brands, with consumers more likely to recommend the brand. Despite recent academic debates on CBAB [147, 148], scholars have yet to explore the mechanism how CSR impacts CBAB. We propose that CSR is a crucial factor in influencing consumer behaviors, including CBAB, as it can create positive organizational perceptions, increasing the likelihood of consumers expressing positive opinions through communicative activities, resulting in increased positive word-of-mouth due to CSR orientation of a particular banking organization.

Our second theoretical contribution emphasizes human emotions' critical role in influencing consumer behavior and psychology. Our research examines human emotions' mediating

**Table 4. Hetrorait-Monotrait Ratio (HTMT).**

|  | ATVL | BRAD | CBAB | CSR | HP |
|---|---|---|---|---|---|
| ATVL |  |  |  |  |  |
| BRAD | 0.642 |  |  |  |  |
| CBAB | 0.397 | 0.621 |  |  |  |
| CSR | 0.587 | 0.665 | 0.562 |  |  |
| HP | 0.612 | 0.794 | 0.663 | 0.664 |  |

**Table 5. Structural analysis.**

| Hypotheses | Estimates | SD | *t-value* | *p*-value | LLCI | ULCI |
|---|---|---|---|---|---|---|
| (CSR→CBAB) | 0.165 | 0.073 | 2.26 | 0.024 | 0.034 | 0.309 |
| (CSR→HP) | 0.415 | 0.068 | 6.12 | 0 | 0.275 | 0.542 |
| (CSR→BRAD) | 0.407 | 0.065 | 6.276 | 0 | 0.276 | 0.522 |
| **Indirect effects** | | | | | | |
| (CSR→HP→CBAB) | 0.130 | 0.039 | 3.337 | 0.001 | 0.066 | 0.219 |
| (CSR→BRAD→CBAB | 0.089 | 0.037 | 2.429 | 0.015 | 0.025 | 0.170 |
| **Conditional effects** | | | | | | |
| CSR*ATVL→HP→CBAB | 0.062 | 0.022 | 2.818 | 0.003 | 0.031 | 0.135 |
| CSR*ATVL→BRAD→CBAB | 0.049 | 0.016 | 3.062 | 0.018 | 0.006 | 0.059 |

role (e.g., HP and BRAD) in explaining how and why CSR influences CBAB in consumers. Though human emotions' importance in consumers' psychology and behavior management has been discussed, most CSR-consumer behavior studies are evaluative (cognitive) [149, 150],

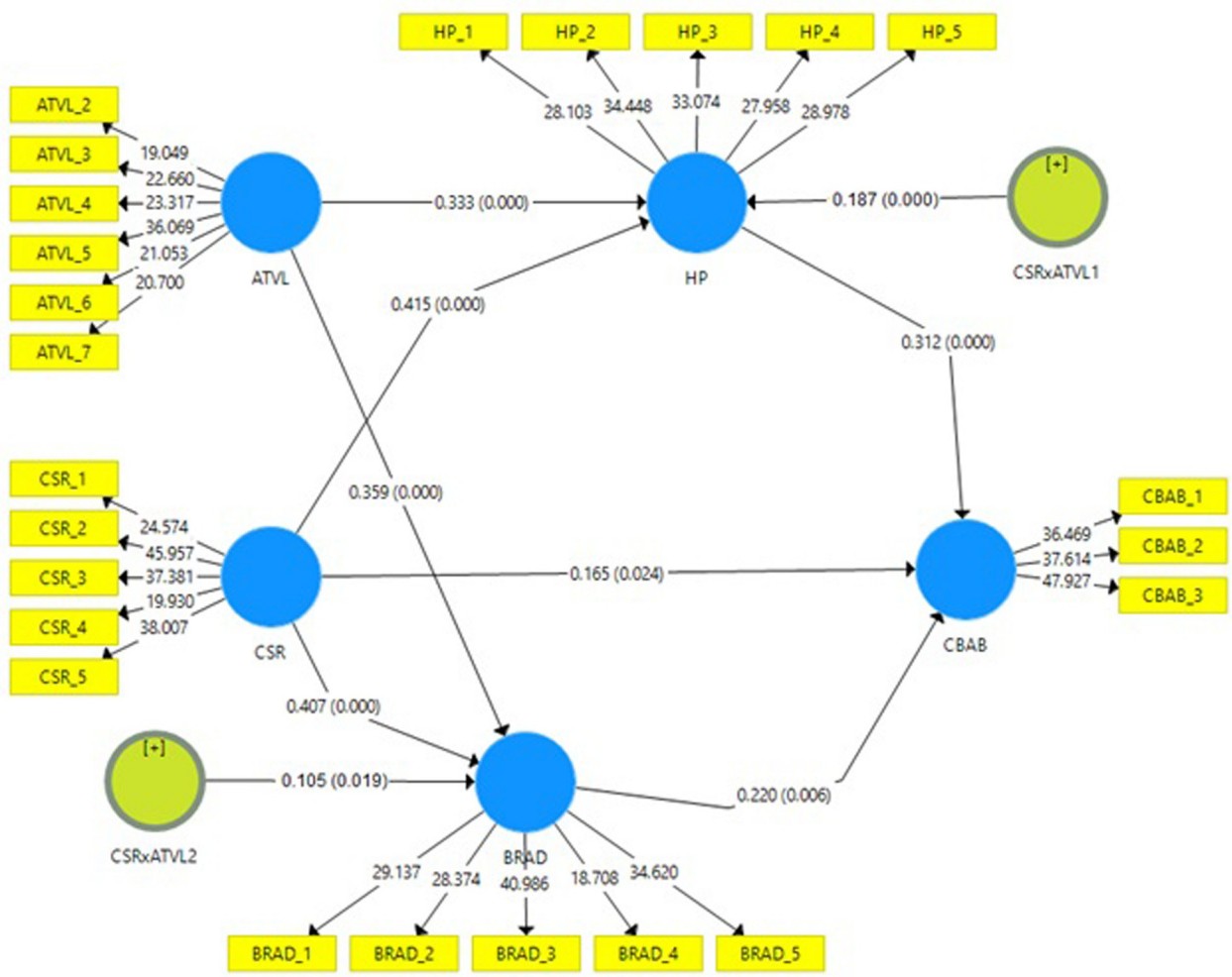

**Fig 2. The structural model showing the hypothesized relationships along with path coefficients.**

**Table 6. Summary of finding.**

| Variables | Key Contributing Factors | Key Implications |
|---|---|---|
| CSR | Environmental Responsibility, Community Outreach | Prioritize these aspects in CSR initiatives<br>Communicate these initiatives effectively through social media platforms<br>Involve customers in community outreach programs |
| HP | Personalized Interaction, Prompt Service Resolution | Enhance customer service experience with personalized interactions<br>Implement quick and efficient problem-solving measures<br>Invest in training staff to provide superior service |
| BRAD | Trustworthiness, Authenticity | Strengthen these aspects in brand image building<br>Maintain transparency in operations<br>Communicate authenticity through consistent brand messaging |
| AVTL | Altruism, Ethical Conduct | Tailor CSR communication toward this consumer group<br>Engage in ethical banking practices<br>Emphasize altruism in brand's mission and values |

neglecting emotions. We introduced HP and BRAD to address this knowledge gap and better understand the complex CSR-CBAB relationships among banking consumers.

Third, our research addresses the traditional CRM model's limitations, criticized for its ineffectiveness in establishing long-term, meaningful relationships with consumers [67, 68]. Traditional CRM focuses on transactional relationships and fails to establish long-term consumer relationships or consider consumer experiences and brand relationships. It also lacks efficiency due to its one-sided communication strategy. We bridge this gap by highlighting social media's potential role as an effective, interactive communication medium to replace the conventional CRM model. By considering social media (a two-way interactive communication forum), we eliminate the issue of one-sided communication in the conventional CRM model. CSR-related communication on social media helps organizations build effective, long-term relationships with consumers.

Lastly, our research emphasizes CBAB's importance in the services industry. CBAB benefits organizations of all sizes and industries. However, in the service industry, having consumer brand ambassadors is crucial, as services often don not provide prior experiences or testing, making personal recommendations even more critical [69–71]. Thus, CBAB is particularly relevant to the services industry. Our study highlights CBAB's crucial role in Pakistan's banking industry, which faces similar competitive convergence challenges as other countries. Consequently, banking organizations can differentiate themselves from competitors by promoting advocacy behavior among consumers. Despite the widespread belief that consumer brand advocates create various benefits for service providers, literature has not adequately focused on its importance. For banking organizations, understanding these detailed insights can guide their CSR strategy. Prioritizing environmental responsibility and community outreach in their CSR initiatives, providing personalized interactions and prompt service resolution, and building a brand image of trustworthiness and authenticity can significantly elevate CBAB. Also, recognizing the consumer group that values altruism and ethical conduct can further optimize their CSR communication and engagement strategy.

## 5.2 Implications for practice

Our research imparts valuable insights into the banking sector, yielding three practical findings. Firstly, it is crucial that banks leverage digital platforms to effectively communicate their CSR initiatives, as this significantly influences consumer advocacy intentions. In the competitive landscape of the banking industry, an effective CSR communication strategy can foster positive consumer associations, amplify brand visibility and recognition, and drive consumers

to recommend the bank. More so, transparency and social responsibility are highly valued by today's consumers and stakeholders, translating into increased consumer loyalty and a healthy bottom line for banking organizations. Hence, an effective social media strategy for CSR activities is instrumental in converting consumers into brand advocates, which is far more influential than traditional marketing messages.

Secondly, banks should consider enhancing their Customer Relationship Management (CRM) model by integrating strategies focusing on eliciting positive emotional responses from consumers, such as happiness and brand admiration. An emotional bond can be far more enduring and impactful than transactional relationships. Social media platforms enable direct, two-way communication, allowing for relationship-building and personalized experiences, leading to stronger emotional connections and, thus, customer loyalty. Therefore, a shift towards an emotion-based perspective could yield substantial benefits for banks.

Lastly, our research highlights the importance of understanding consumers as not just loyal customers but also as potential brand advocates who can defend the brand against criticism. Banks should, therefore, strive to make their customers feel valued and appreciated, which can generate positive word-of-mouth and help the brand stand out among competitors. Engaging consumers emotionally through CSR activities is particularly crucial in the service industry, where human dependencies on service delivery surpass those of physical goods.

### 5.3 Limitations and suggestions for future

Although our research significantly enriches the existing theoretical debate in a unique way and, at the same time, offers different practical insights, it is still not without limitations. Our first potential limitation lies with geographical concentration. We collected data from consumers residing in Lahore. Though important, we feel this geographical concentration may not present the whole country's consumers' perspective. We, therefore, suggest including more cities in future studies. Our second limitation lies with the study design. We used a cross-sectional data collection strategy due to time and other constraints. Though this data collection strategy has been employed by many researchers previously, we feel for causal relations, a better strategy would be to opt for a longitudinal study design. Another limitation lies with single industry interventions. In this study, we only considered the banking sector. Other service segments, like hospitality, also face the issue of competitive convergence. Therefore, extending our framework to other industries or producing some comparative analysis will be helpful in the future. Moreover, while our study highlights the effectiveness of social media in promoting CSR initiatives, it does not directly compare this approach with traditional, offline methods of CSR communication. The relative impact of these two communication channels might differ, and a comprehensive comparison might yield significant insights into CSR strategy. We recognize this as a limitation of our study. Future studies might aim to establish a comparative analysis between social media CSR and offline CSR communication methods, thereby expanding upon our findings and providing a further understanding of CSR communication effectiveness in various channels.

### 6. Conclusion

In summary, social media has transformed the business landscape by offering different benefits, including the ability to engage in dialogic communication with consumers. Social media grants businesses unparalleled access to consumer feedback, fostering agility in product and service development. Furthermore, it provides an opportunity to build relationships with consumers, significantly affecting consumer loyalty. In the digital era, consumers extensively utilize social networking platforms for information, awareness, knowledge, and entertainment. A

social media presence allows businesses to interact with consumers in real-time. As sustainability issues gain prominence, consumers increasingly favor socially responsible organizations. Banking organizations can leverage CSR programs by engaging with their consumers. By communicating their CSR initiatives, banking organizations can establish a positive reputation and demonstrate commitment to positively impacting society and ecosystems, attracting more consumers and enhancing loyalty.

Modern businesses need to acknowledge the potential of social media as an interactive communication medium capable of supplanting the conventional one-way CRM communication model. Social media enables direct and prompt interaction with consumers, offering a more personalized experience and facilitating a better understanding of consumer expectations. This fosters stronger relationships with consumers. Social media's customizable, dialogic, and interactive features empower businesses to emotionally connect with their consumers, leading to a more profound, meaningful bond. Ultimately, a banking organization that converts consumers into brand advocates through CSR strategies and communicates these initiatives via social media can secure a stable competitive edge over competitors.

## 7. Key recommendations

➢ Utilize social media for CSR outreach: The banking industry should harness the power of social media platforms to disseminate CSR efforts, as this strategy has proven to bolster consumer advocacy intentions. These digital platforms offer real-time, interactive communication, helping foster trust and a sense of community around the brand.

➢ Emphasize emotional connections: Banks should transition towards an upgraded CRM model that underscores building emotion-centric relationships with consumers, moving away from solely transaction-based ones. This could be realized by comprehending and catering to consumers' emotional requirements, crafting personalized experiences, and evoking positive emotions via swift service solutions and meaningful exchanges.

➢ Involve consumers in CSR endeavors: Banks should strive to engage consumers in their CSR endeavors, as this engagement can amplify consumer emotional involvement and yield higher happiness and brand admiration levels. Such engagement could be made possible through social media platforms, where consumers can actively participate in the bank's CSR initiatives.

➢ Encourage altruism: The banking sector should recognize and value consumers who exhibit altruistic values and ethical behavior. These consumers are more likely to partake in advocacy behaviors, and hence, could be the focus of specific CSR endeavors that resonate with their values.

➢ Prioritize environmental responsibility and community engagement: In executing their CSR plans, banks should particularly focus on environmental responsibility and community outreach. These facets of CSR have shown to leave a significant impact on consumer advocacy behavior.

➢ Promote transparency: Open and public CSR communications on social media can trigger emotional ties with consumers, paving the way for loyalty and recommendations. As such, banks should prioritize transparency in their CSR disclosures.

➢ Foster consumer brand ambassadors: Banks should strive not only for consumer loyalty but also to transform satisfied customers into brand ambassadors. These brand ambassadors can serve as influential marketing assets, promoting and shielding the brand against

criticism, thereby bolstering the bank's reputation. This can be achieved by ensuring consumers feel acknowledged and cherished.

## Supporting information

**S1 File.**
(TXT)

## Author Contributions

**Conceptualization:** Yang Liu, Rana Tahir Naveed, Sara Kanwal.

**Data curation:** Sara Kanwal.

**Formal analysis:** Yang Liu, Rana Tahir Naveed.

**Methodology:** Yang Liu, Rana Tahir Naveed, Muhammad Tahir Khan.

**Software:** Muhammad Tahir Khan, Wei Lan.

**Supervision:** Ali F. Dalain.

**Writing – original draft:** Wei Lan.

**Writing – review & editing:** Yang Liu, Sara Kanwal, Muhammad Tahir Khan, Ali F. Dalain, Wei Lan.

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
