## [Decision Letter · Decision Letter 0]

3 Jul 2023

PONE-D-23-17895Psychology in Action: Social Media communication, CSR, and Consumer Behavior Management in BankingPLOS ONE

Dear Dr. Naveed,

Thank you for submitting your manuscript to PLOS ONE. After careful consideration, we feel that it has merit but does not fully meet PLOS ONE’s publication criteria as it currently stands. Therefore, we invite you to submit a revised version of the manuscript that addresses the points raised during the review process.

We look forward to receiving your revised manuscript.

Kind regards,

Kittisak Jermsittiparsert, Ph.D.

Academic Editor

PLOS ONE

We will update your Data Availability statement to reflect the information you provide in your cover letter."

5. Please include a copy of Table 5 which you refer to in your text on page 14.

Additional Editor Comments (if provided):

This article has an interesting point. All reviewers have a positive opinion on it and suggest minor improvements. I, as the editor, have decided that the author must improve this article strictly based on all those suggestions.

Reviewers' comments:

Reviewer's Responses to Questions

**Comments to the Author**

1. Is the manuscript technically sound, and do the data support the conclusions?

Reviewer #1: Yes

Reviewer #2: Yes

Reviewer #3: Yes

2. Has the statistical analysis been performed appropriately and rigorously? 

Reviewer #1: Yes

Reviewer #2: Yes

Reviewer #3: Yes

3. Have the authors made all data underlying the findings in their manuscript fully available?

Reviewer #1: No

Reviewer #2: Yes

Reviewer #3: Yes

4. Is the manuscript presented in an intelligible fashion and written in standard English?

Reviewer #1: Yes

Reviewer #2: Yes

Reviewer #3: Yes

5. Review Comments to the Author

Reviewer #1: - A large part of this article discusses the importance of this study and the importance of the studied variables such as brand advocacy, CSR activities, which are already accepted. More should be discussed about the findings.

- Please explain why Consumer Brand Advocacy Behavior is shortened as CODB. (why the D?)

- More should be explained about the sampling methods. How were the 600 questionnaires distributed? How did you select the ATM areas or the banks? What is the demographic data of the final 391 responses?

- A sample statement of CSR item "When I see CSR related communication on my bank…" : This might lead to a bias for the judgment of one's own bank for he/she is already the customer and tends to view own's bank favorably.

- What is the final number of valid responses? (409 filled questionnaires minus 19 invalid cases should yield 390 responses, not 391 as identified in the article. Also, how do the authors justify the sample size (390 or 391) as sufficient? Why didn't the researchers make it into at least 400?

- The discussion and implication parts talk mostly about general ideas. This article should be much more valuable if the authors can point out more in detail, such as which factors of each variables contribute most to CODB.

- If possible, a summary table should make the conclusion much more understandable.

- For 5.2 Implications for practice, this article should recommend practical approaches for organizations, especially banks, how to improve their CSR communication/activities on social media. For example. focusing on activities that increase customer happiness and brand admiration based on certain altruistic values.

- The authors might've gone too far to argue that social media CSR can supplant traditional/offline CSR because this study doesn't compare the results between two approaches, which should be a subject of future research.

- Selecting banks as the subject of this study might also present another limitation in the generalization of the study, because banking services are not something that consumers can easily recommend another person to use, just like household goods. So the authors might want to explain the meaning of "CODB" in the context of banking industry.

Reviewer #2: Recommendations to revise the research article structure according to the following issues:

Abstract

Introduction

Research Objectives or Research Hypotheses

Literature Review

Methodology

Results

Discussions and Conclusions

Recommendations (if applicable)

Reviewer #3: The abstract provides a concise overview of the research topic and outlines the objectives and key findings of the study. The introduction provides a good overview of the importance of consumer brand advocacy behavior (CODB) in the banking sector and its impact on trust, reputation, and market position. It also highlights the role of CODB in reducing consumer acquisition costs and navigating crises. However, there are a few suggestions to enhance clarity and coherence: Expand on the link between CSR initiatives and CODB: The introduction briefly mentions fostering CODB through CSR initiatives as a means to secure a competitive advantage. It would be helpful to provide a more explicit explanation of how CSR initiatives can influence CODB. This could include examples of CSR activities in the banking sector that have been shown to positively impact consumer advocacy. the result and conclusion is

clear and effectively conveys the benefits of social media in engaging consumers and promoting CSR initiatives.

6. PLOS authors have the option to publish the peer review history of their article (what does this mean?). If published, this will include your full peer review and any attached files.

Reviewer #1: **Yes: **Assistant Professor Warat Karuchit, Ph.D

Reviewer #2: No

Reviewer #3: **Yes: **Associate Professor. Sergeant Major Third Class. Dr. Nath Amornpinyo

---

## [Author Response · Author response to Decision Letter 0]

12 Jul 2023

Response: We have formatted our manuscript as per the format specified by PLOS ONE. 

We will update your Data Availability statement to reflect the information you provide in your cover letter."

Response: Thanks for this clarification. The raw data file has been uploaded during the revision submission. Best Regards 

Response: Thank you for your guidance regarding the placement of our ethics statement. We've now moved the statement to the Methods section, ensuring its removal from other sections. We understand the importance of having it within the manuscript itself and apologize for any initial confusion.

5. Please include a copy of Table 5 which you refer to in your text on page 14.

Response: Thank you for this keen observation. Actually there was an error in reporting table numbers. In the earlier version Table 5 was mistakenly labeled as Table 6. Nonetheless, we have rectified this error during the revision. Thanks again for this keen observation

Additional Editor Comments (if provided):

This article has an interesting point. All reviewers have a positive opinion on it and suggest minor improvements. I, as the editor, have decided that the author must improve this article strictly based on all those suggestions.

Reviewers' comments:

Reviewer's Responses to Questions

Comments to the Author

1. Is the manuscript technically sound, and do the data support the conclusions?

Reviewer #1: Yes

Reviewer #2: Yes

Reviewer #3: Yes

2. Has the statistical analysis been performed appropriately and rigorously?

Reviewer #1: Yes

Reviewer #2: Yes

Reviewer #3: Yes

3. Have the authors made all data underlying the findings in their manuscript fully available?

Reviewer #1: No

Reviewer #2: Yes

Reviewer #3: Yes

Response: We have uploaded the raw data file during the revision submission stage 

4. Is the manuscript presented in an intelligible fashion and written in standard English?

Reviewer #1: Yes

Reviewer #2: Yes

Reviewer #3: Yes

Response: The manuscript has been cross-checked for any grammar related error. 

5. Review Comments to the Author

Reviewer #1: - A large part of this article discusses the importance of this study and the importance of the studied variables such as brand advocacy, CSR activities, which are already accepted. More should be discussed about the findings.

Response: Thank you for your constructive comments and insights. We appreciate your input and understand your concerns regarding the need for a deeper discussion on our study findings. We agree that while the importance of the variables studied, such as brand advocacy and CSR activities, are generally accepted, the crux of our contribution lies in our specific results. In light of your feedback, we have expanded our discussion on the findings. Specifically, we have added the following text in the revised manuscript for your kind perusal. 

Specifically, our study tends to highlight the profound influence of CSR activities on consumer advocacy because the past literature suggests that CSR can influence consumer behavior significantly (71, 72). When brands not only engage in, but also effectively communicate their CSR initiatives through social media, it stimulates consumers to advocate and defend the brand online, thereby enhancing the brand's reputation and trustworthiness. Furthermore, our research tends to demonstrate the crucial mediating role of HP and BRAD in the relationship between CSR and CBAB. CSR initiatives that promote HP and BRAD, which subsequently amplify CBAB. This suggest that by strategically aligning CSR activities to generate positive emotions among consumers, brands can substantially enhance the advocacy intentions of consumers.

Similarly, the proposed findings of this research will be helpful for brands in acknowledging the role of ATVL as a significant moderating factor in the CSR-CBAB mechanism. We believe that individuals with strong altruis are more likely to express CBAB when brands actively participate in meaningful CSR activities. Consequently, brands can design their CSR strategies and communications to cater to consumers' altruistic tendencies, fostering stronger consumer-brand relationships in the process which ultimately foster their advocacy intentions. More specifically, in the context of the banking sector, our research tends to indicate that CSR-induced CBAB significantly reinforces trust and reputation. In an industry with limited differentiating opportunities, CSR initiatives, by enhancing CBAB, appear to have substantial implications for client acquisition and retention. Furthermore, we propose that strong CBAB can provide a supportive base during turbulent times, aiding banks in crisis management. Moreover, using social media platforms to disseminate information about CSR initiatives greatly enhances bank perceptions and advocacy. Hence, the proposed findings of this research will be helpful for brands to recognize the value of these platforms as they provide a perfect medium for two-way dialogue with consumers, thereby facilitating the building of stronger relationships.

We hope that these revisions address your concerns. We appreciate your invaluable input in strengthening our manuscript and look forward to any further suggestions you may have.

- Please explain why Consumer Brand Advocacy Behavior is shortened as CODB. (why the D?)

Response: Apology for creating this unintended confusion, the term CODB has now been replaced with CBAB throughout the revised text. Thanks for the kind feedback. 

- More should be explained about the sampling methods. How were the 600 questionnaires distributed? How did you select the ATM areas or the banks? 

Response: Thank you for the above feedback. The 600 questionnaires were distributed directly to customers as they exited bank branches or visited ATM service areas in Lahore. We adopted a face-to-face approach to facilitate any clarifications the participants might need and to ensure the quality of the data collected. This method also offered the added benefit of not disrupting the delivery of banking services. As for the selection of banks and ATM areas, the banks were chosen based on their market presence, the comprehensiveness of their CSR initiatives, and their use of social media for customer engagement. We aimed to ensure a diverse mix, so we selected four conventional banks and two Islamic banks, thereby covering the primary banking systems in Pakistan. The ATM locations were chosen strategically. We identified areas with high foot traffic within the operational vicinities of the selected banks. This was done to increase the likelihood of interacting with a diverse group of customers who engage with the banks' services frequently. Regards 

What is the demographic data of the final 391 responses?

Response: The demographic detail have been added in the revised manuscript as below

Out of the total participants, males accounted for a majority share of 67.18% (262 individuals), whereas females constituted 32.82% (128 individuals). In terms of age demographics, the respondents were divided into five different age groups. The age group with the highest representation was the 26-35 years old bracket, comprising 36.92% of the total respondents. Following this, the age groups of 36-45 and 46-50 years represented 20.51% and 17.70% respectively. The younger age group of 18-25 years represented a lesser portion, contributing to 11.79%, while the respondents aged above 50 years accounted for the remaining 13.08%. Regarding the educational background of respondents, those holding a graduate degree made up 37.95% (148 individuals), and 35.13% (137 individuals) possessed a master's degree. Respondents with an intermediate level of education constituted a smaller segment of 12.56% (49 individuals). Those who had an education beyond the master's degree represented 14.36% (56 individuals) of the total respondents.

- A sample statement of CSR item "When I see CSR related communication on my bank…" : This might lead to a bias for the judgment of one's own bank for he/she is already the customer and tends to view own's bank favorably.

Response: Thank you for your insightful comment. The statement, "When I see CSR-related communication of my bank on a social networking website, I feel that this bank is socially responsible," is intended to assess the participant's perception of their bank's CSR actions as communicated through social media platforms. We understand your concern regarding the potential bias due to participants' pre-existing favorable view of their own banks. However, our research specifically investigates the impact of CSR communication via social media on reshaping or reinforcing customers' impressions about their bank's commitment to social responsibility.

Your comment underscores a vital facet of research regarding acknowledging and addressing inherent biases in survey studies. Yet, the kind of bias you have identified is inherent in almost all customer perception surveys, which naturally draw responses from customers who have already opted for the brand or service in focus. Additionally, we would like to highlight that the bias you are referring to can manifest in two ways. Customers may indeed have a more positive perception of their bank, but they might also maintain higher expectations due to their direct relationship and experiences with the bank. We maintain that any potential bias does not negate the findings but rather offers a context for interpreting the results. We trust this clarifies our position and addresses your query. Regards 

- What is the final number of valid responses? (409 filled questionnaires minus 19 invalid cases should yield 390 responses, not 391 as identified in the article. Also, how do the authors justify the sample size (390 or 391) as sufficient? Why didn't the researchers make it into at least 400?

Response: We appreciate your query about the sample size in our study. Our sample size determination relied on the A-priori sample size calculator by Daniel (2010), a tool highly regarded in social and behavioral sciences surveys, and particularly useful when applying multivariate data analysis techniques such as CB-SEM and PLS-SEM. This tool bases its calculations on several factors, encompassing the count of latent and observed variables, the probability level, and the effect size, thus guiding towards an appropriate sample size. For our study, the suggested sample size was 346 individuals, according to the parameters we provided. 

We collected responses from 409 participants (which became 390 after data cleaning and the exclusion of outliers), significantly surpassing the proposed sample size. This larger number strengthens the reliability and robustness of our study findings. The count of participants should be perceived as a minimum limit, with any count above the recommended value serving as an additional safeguard. While rounding off to an even number like 400 might have been more conventional, it doesn't inherently offer statistical benefits over a count like 390. As such, even though we acknowledge your suggestion, we maintain that our sample size is adequately substantial for our study and aligns with the calculated recommendation.

- The discussion and implication parts talk mostly about general ideas. This article should be much more valuable if the authors can point out more in detail, such as which factors of each variables contribute most to CODB.

Response: Thanks again for the kind feedback. We have revised our discussion and implication part while revising our manuscript. Kindly see the newly added text (yellow highlighted in the revised discussion part). Best Regards 

- If possible, a summary table should make the conclusion much more understandable.

Response: Table 6 has been added in the discussion section for your kind perusal. Regards 

- For 5.2 Implications for practice, this article should recommend practical approaches for organizations, especially banks, how to improve their CSR communication/activities on social media. For example, focusing on activities that increase customer happiness and brand admiration based on certain altruistic values.

Response: Thanks again for the feedback. We have revised our implication for practice section. Hopefully, the modified implications will satisfy you. Best Regards 

- The authors might've gone too far to argue that social media CSR can supplant traditional/offline CSR because this study doesn't compare the results between two approaches, which should be a subject of future research.

Response: We have added the following text in the revised limitation section 

Moreover, while our study highlights the effectiveness of social media in promoting CSR initiatives, it does not directly compare this approach with traditional, offline methods of CSR communication. The relative impact of these two communication channels might differ, and a comprehensive comparison might yield significant insights into CSR strategy. We recognize this as a limitation of our study. Future studies might aim to establish a comparative analysis between social media CSR and offline CSR communication methods, thereby expanding upon our findings and providing a further understanding of CSR communication effectiveness in various channels.

- Selecting banks as the subject of this study might also present another limitation in the generalization of the study, because banking services are not something that consumers can easily recommend another person to use, just like household goods. So the authors might want to explain the meaning of "CODB" in the context of banking industry.

Response: Certainly, the nature of banking services differs from conventional household goods, as they may not be recommended among consumers as readily. Nevertheless, our decision to focus on banks was not arbitrary. Banks, by their inherent attributes, necessitate a more profound level of trust, commitment, and loyalty from their customers than other sectors. These traits make the banking sector an essential and appropriate platform to investigate the influence of CSR communication on CBAB. Within the context of the banking sector, CBAB is perceived as a propensity of banking customers to actively participate in, advocate, and disseminate positive experiences regarding their chosen banks. This behavior is crucial for services industries including banking industry as positive referrals from reliable sources can dramatically influence an individual's bank selection. Therefore, although banking services may not be as casually recommended as household goods, the impact of recommendations in banking is considerably more substantial. However, we recognize the potential limitations in extrapolating our findings to different industries. Future research could apply the conclusions of this study to other sectors, possibly revealing varying impacts of CSR communication based on the specificities of each industry. This expansion would provide a more in-depth understanding of CSR's role and effect across various industrial contexts

Reviewer #2: Recommendations to revise the research article structure according to the following issues:

Abstract

Introduction

Research Objectives or Research Hypotheses

Literature Review

Methodology

Results

Discussions and Conclusions

Recommendations (if applicable)

Response: Thanks for the above feedback. We have divided our revised manuscript in the above suggested sections. Thanks again for the feedback. 

Reviewer #3: The abstract provides a concise overview of the research topic and outlines the objectives and key findings of the study. The introduction provides a good overview of the importance of consumer brand advocacy behavior (CODB) in the banking sector and its impact on trust, reputation, and market position. It also highlights the role of CODB in reducing consumer acquisition costs and navigating crises. However, there are a few suggestions to enhance clarity and coherence: Expand on the link between CSR initiatives and CODB: The introduction briefly mentions fostering CODB through CSR initiatives as a means to secure a competitive advantage. It would be helpful to provide a more explicit explanation of how CSR initiatives can influence CODB. This could include examples of CSR activities in the banking sector that have been shown to positively impact consumer advocacy.

Response: Thank you for your valuable feedback on our manuscript. In response to your comment, we have included examples of successful CSR activities in the banking industry. Furthermore, we have also emphasized the importance of CSR initiatives in building consumer trust and loyalty from the perspective of banking sector. Specifically, we have added/modified the following text in the revised manuscript. 

In the context of the banking industry, banks which aptly articulate and fulfill their CSR obligations receive positive customer evaluation. Initiatives such as backing renewable energy projects, aiding local enterprises, or introducing financial literacy programs gain considerable favor with consumers. Successful examples of such practices include banks like Bank of America, which pledged $300 billion towards low-carbon business efforts, and Standard Chartered, renowned for its numerous community engagement initiatives. Highlighting a specific example from Pakistan, Habib Bank Limited (HBL) has demonstrated exceptional success with its comprehensive CSR programs. HBL has made significant inroads in endorsing education, public health, and environmental sustainability, thereby positively influencing their client base. In a similar vein, United Bank Limited (UBL) has put forth considerable CSR contributions, with an emphasis on community development and disaster relief. These CSR endeavors not only solidify the banks' image but also motivate customers to champion these brands, leading to enhanced loyalty and customer retention. Hence, by fostering CBAB through CSR initiatives, banks can secure a competitive advantage and drive long-term success

Comment: The result and conclusion is

clear and effectively conveys the benefits of social media in engaging consumers and promoting CSR initiatives.

Response: Thank you for your insightful and positive feedback. We appreciate your recognition of the clarity and effectiveness of our results and conclusion. Best Regards

---

## [Editor Report · Decision Letter 1]

17 Jul 2023

Psychology in Action: Social Media communication, CSR, and Consumer Behavior Management in Banking

PONE-D-23-17895R1

Dear Dr. Naveed,

We’re pleased to inform you that your manuscript has been judged scientifically suitable for publication and will be formally accepted for publication once it meets all outstanding technical requirements.

Kind regards,

Kittisak Jermsittiparsert, Ph.D.

Academic Editor

PLOS ONE

Additional Editor Comments (optional):

Thank you for the interesting paper and improving it based on the reviewers' suggestions to satisfactory.
---

## [Editor Report · Acceptance letter]

8 Aug 2023

PONE-D-23-17895R1 

Psychology in action: Social media communication, CSR, and consumer behavior management in banking 

Dear Dr. Naveed:

I'm pleased to inform you that your manuscript has been deemed suitable for publication in PLOS ONE. Congratulations! Your manuscript is now with our production department. 

Kind regards, 

on behalf of

Professor Kittisak Jermsittiparsert 

Academic Editor

PLOS ONE